# Management of *de novo* metastatic hormone-sensitive prostate cancer: A comprehensive report of a single-center experience

Sunny Guin[1], Bobby K. Liaw[2], Tomi Jun[2], Kristin Ayers[1], Bonny Patel[1], Timmy O'Connell[1], Matthew Deitz[1], Michael Klein[1], Tommy Mullaney[1], Tony Prentice[1], Scott Newman[1], Marc Fink[1], Xiang Zhou[1], Eric E. Schadt[1,2], Rong Chen[1,2]*, William K. Oh[1,2]*

1 Sema4, Stamford, CT, United States of America, 2 Mount Sinai Health System, New York, NY, United States of America

⊛ These authors contributed equally to this work.

* Rong.chen@sema4.com (RC); William.oh@sema4.com (WKO)

**Data Availability Statement:** Data cannot be shared publicly because of IRB restrictions and patient confidentiality. Data are available from the Sema4 Institutional Data Access / Ethics

# Abstract

## Background

Upfront docetaxel or novel hormonal agents (NHA) such as abiraterone and enzalutamide have become the standard of care for metastatic hormone sensitive prostate cancer (mHSPC). We evaluated real-world management of patients treated with these agents at a single center.

## Patients and methods

Patients with *de novo* mHSPC treated with upfront docetaxel or an NHA between January 2014 and April 2019 at Mount Sinai Health System were included. We evaluated time to next treatment (TTNT), PSA progression free survival (PFS) and overall survival (OS) after initial treatment with these drugs. Kaplan Meier method and multivariable Cox proportional hazards models were used for analysis. We additionally assessed the prognostic value of post-treatment PSA.

## Results

We identified 94 *de novo* mHSPC patients; 52 and 42 treated with upfront docetaxel and NHAs, respectively. NHAs were associated with a median TTNT of 20.7 months compared to 10.1 months with docetaxel (log-rank p = 0.023). We also observed median PSA PFS of 19 months for NHAs and 13.2 months for docetaxel (p = 0.069). However, OS between the two treatment groups was unchanged. Among docetaxel treated patients, TTNT was shorter among those with high metastasis burden (9.63 vs 25.5 months, p = 0.026) which was not observed among NHA treated patients (25.1 vs 20.7 months, p = 0.79). Regardless of treatment, lower post-treatment PSA levels were associated with improved TTNT (58.95 vs. 11.57 vs. 9.4 months for PSA ≤0.2, 0.2–0.4, >0.4ng/ml, respectively; p<0.001)

Committee (contact via irb-rnd@sema4.com) for researchers who meet the criteria for access to confidential data.

**Funding:** The author(s) received no specific funding for this work.

**Competing interests:** The authors have declared that no competing interests exist.

## Conclusion

Real world data demonstrated a shorter duration of treatment with docetaxel than NHAs, reflecting the time-limited nature of docetaxel regimens compared to the treat-till-progression approach of NHAs. While TTNT was generally longer for NHAs than docetaxel, some docetaxel-treated patients achieved significant periods of time off treatment. In addition, the depth of PSA response following combination treatment may hold prognostic value for mHSPC outcomes.

## Background

The standard of care for metastatic hormone sensitive prostate cancer (mHSPC) continues to quickly evolve. While the use of androgen deprivation therapy (ADT) remains foundational, the earlier incorporation of advanced therapeutic agents previously reserved for castration resistant disease has become routine in patients with mHSPC. The CHAARTED (2015) and LATITUDE (2017) trials led to the incorporation of docetaxel and abiraterone, respectively, as standard of care options in combination with ADT for mHSPC; each demonstrating significant overall survival (OS) advantages over ADT alone [1, 2]. Enzalutamide and apalutamide approvals followed in 2019, based on positive data from ARCHES, ENZAMET, and TITAN trials [3–5].

Data to guide the optimal selection among these 1st line treatment regimens for mHSPC are limited. Meta-analyses comparing clinical trials of upfront docetaxel versus novel hormonal agents (NHA) in mHSPC have not shown any overall or progression-free survival benefit for either class of medication over the other [6–8]. To fill this gap, we performed a retrospective analysis of *de novo* mHSPC patients treated with docetaxel or novel hormonal agents (NHA) within the Mount Sinai Health System. We developed an automated oncology data retrieval and curation platform, and comprehensively extracted clinical features, outcomes, toxicities, and treatment patterns from the electronic medical record (EMR) to characterize and compare the clinical outcomes of patients treated with either upfront docetaxel or NHA for *de novo* mHSPC.

## Methods

### Data source

Patients with prostate cancer were identified from the Mount Sinai Hospital at New York City data warehouse. Original EMR data from the hospital data warehouse were processed using the Sema4 Centrellis™ platform, consisting of automated abstraction engine, curation platform, patient dashboard, and cohort builder. Along with information from structured data fields, data from unstructured clinical notes were automatically abstracted, curated and integrated into structured fields. Cancer diagnosis metastases, medication, treatment, PSA values and other related data elements were stored in the database

### Patient cohort identification

We identified prostate cancer patients diagnosed between January 2014 and April 2019 to capture the time frame when docetaxel and NHA were approved for mHSPC. 11,358 patients were assigned with at least one prostate cancer related ICD codes (185 in ICD-9 code and C61

in ICD-10 code) between January 1st, 2014 and April 30th, 2019 in the EMR system. However, not all patients with prostate cancer ICD codes assigned have confirmed prostate cancer. In addition, for patients diagnosed prior to visiting Mount Sinai hospital, their first time ICD code recorded in the EMR system didn't always match the date of diagnosis. To correctly identify patients with pathologically confirmed cancer diagnosis dates, we used the automated oncology data retrieval and curation platform developed by Sema4 as demonstrated in S1A Fig. Both structured information and extracted data elements from unstructured clinical note using heuristic rules were used to assign cancer diagnosis dates as demonstrated in S1B Fig [9]. Patients with their extracted cancer diagnosis dates and other diagnosis related data elements, such as stage, histology, TNM, were classified into three groups using a stratification algorithm. Selected patients' data were manually reviewed by domain experts to evaluate the quality of the results We excluded patients diagnosed after April 2019 to ensure at least 12 months of follow-up. We defined *de novo* metastatic disease as clinical documentation of metastatic disease within 3 months of the diagnosis date. Upfront docetaxel and NHA use were determined by chart review. After the cohorts were identified, chart review was carried out to confirm duration of upfront docetaxel or NHA therapy, adverse events, identification of 2nd line therapy, and start date of 2nd line therapy. Additional clinical and demographic variables such as PSA values, age, sex, race, performance status, Gleason score, and metastasis burden at time of diagnosis were also evaluated. This study was approved by the Mount Sinai Institutional Review Board (Protocol number: STUDY-21-00442). All experimental protocol for involving human data is in accordance with the Mount Sinai Health System (institutional) and Declaration of Helsinki guidelines. Participant informed consent was obtained in writing before they were included in the database for analysis and the study was approved by the Mount Sinai Health System ethics committee as mentioned above.

## Time of next treatment and overall survival analysis

Time to next treatment (TTNT) was evaluated for this study, defined as time from initiation of either docetaxel or NHA to the next line of treatment. For the purposes of this metric, any change or addition of systemic therapy–including chemotherapy, NHA, immunotherapy, and clinical trials–was considered a failure of therapy, regardless of the reason for the change (e.g. progression, toxicity, patient preference, physician discretion). Patients not known to receive a 2nd line treatment were censored at the time of their last follow-up. TTNT was represented as Kaplan-Meier curves as detailed under *Statistical Analysis*. Overall survival (OS) was evaluated for this study, defined as time from diagnosis to death (event) or last follow-up (censor). OS was represented as Kaplan-Meier curves as detailed under *Statistical Analysis*.

## PSA progression free survival (PFS) analysis

PSA PFS was defined as the time from initiation of therapy to first PSA increase $\geq 25\%$ and $\geq 2$ ng/mL above that patient's nadir, confirmed by a second value at least 3 weeks later, per PCWG3 criteria [10]. Patients who were started on 2nd line therapy before criteria for PSA progression were met were excluded from this analysis. Patients who did not have PSA progression and who had not started 2nd line therapy were censored at the time of their last follow-up. PSA PFS was represented as Kaplan-Meier curves as detailed under *Statistical Analysis*.

## Metastasis burden and time of next treatment

High metastasis burden was defined as visceral metastases or $\geq 4$ bone lesions with $\geq 1$ beyond the vertebral bodies and pelvis, per the CHAARTED trial [2]. Patients were assigned: high,

low, or uncertain metastatic burden based on chart review by an oncologist. Differences in TTNT between patient with high and low burden disease were compared in the 1. overall mHSPC cohort; 2. docetaxel treated cohort; and 3. NHA treated cohort. We further compared TTNT between patients treated with docetaxel or NHA, stratified by metastasis burden. TTNT were represented as Kaplan-Meier curves as detailed under *Statistical Analysis*.

### PSA nadir and time to next treatment

The post-treatment PSA in the docetaxel cohort was defined as the PSA value following completion of at least 4 cycles of docetaxel therapy. Patients who started 2nd line treatment prior to finishing 4 cycles of docetaxel (e.g. due to toxicity or refractory disease) were excluded from this analysis. For the NHA cohort, post-treatment PSA value was recorded after patients had received at least 7 months of ADT and at least 3 months of NHA. Patient were categorized in the docetaxel and NHA cohorts into three separate groups based on post-treatment PSA values: $\leq 0.2$, $0.2$–$4.0$, or $\geq 4.0$ ng/mL. We compared TTNT according to post-treatment PSA within each treatment cohort.

### Statistical analysis

Failure-free survival and PSA progression free survival data were presented as Kaplan-Meier curves. Survival curves were created with R (version 3.5.0; http://www.R-project.org/.) using the survival package (version 2.44–1; https://cran.r-project.org/web/packages/survival/index. html) and report the log rank p-value according to default parameters. We examined potential confounding variables that may also be associated with clinical outcome using single variable and multivariable Cox proportional hazards regression, reporting the hazard ratios and the Wald test statistic p-value for each variable. PSA values were log transformed due to the heavy right tail, and 2 individuals with missing values were assigned to the median value. Patients with unknown metastasis burden status were excluded in the Cox regression models.

## Results

### Patient cohort

Total 5725 patients were identified by the platform with pathologically confirmed diagnosis dates for prostate cancer between January 1, 2014 to April 30, 2019. Of these, 240 were determined to have had *de novo* metastatic disease, based on clinical documentation of metastatic disease within 3 months of the diagnosis date. The diagnosis date, metastatic status, and date of metastasis detection were based on terms extracted from clinical notes and pathology reports via the abstraction engine described above. We then identified patients who had been treated with docetaxel (N = 67) or an NHA (abiraterone or enzalutamide) (N = 99) at any time. Finally, we identified those who had been treated with upfront docetaxel (N = 52) or an NHA (N = 42), defined as treatment within 7 months of diagnosis in conjunction with ADT and in the absence of any prior systemic treatment for prostate cancer.

The two groups of *de novo* mHSPC patients had no significant differences in age, race/ethnicity, PSA at diagnosis, Gleason score, ECOG performance status, and metastatic burden. Table 1 report the standardized mean difference (SMD) value. Variables with SMD<0.1 are considered well matched between the two groups. The docetaxel cohort has a higher frequency of patients with a Gleason score of 10 compared to NHA cohort (N = 12 versus N = 1). However, when combining Gleason scores for aggressive disease (9 and 10), the cohorts are well matched with frequencies of 57.7% and 57.1% for the docetaxel and NHA cohorts, respectively. We further evaluated the high burden patients for visceral metastasis, 6 patients with

**Table 1. Table of co-variants for the docetaxel and NHA cohorts.**

| | NHA | Docetaxel | p-value | SMD |
|---|---|---|---|---|
| | 42 | 52 | | |
| Age Diagnosis (mean (SD)) | 67.31 (8.89) | 64.87 (9.47) | 0.204 | 0.266 |
| Age Category | | | 1 | 0.011 |
| 65 and Over | 24 (57.1) | 30 (57.7) | | |
| Under 65 | 18 (42.9) | 22 (42.3) | | |
| PSA (mean (SD)) | 1166.05 (3456.13) | 739.82 (1423.99) | 0.428 | 0.161 |
| log10PSA (mean (SD)) | 2.13 (0.91) | 2.21 (0.85) | 0.664 | 0.091 |
| ECOG Score (%) | | | 0.493 | 0.248 |
| 0 | 16 (38.1) | 25 (48.1) | | |
| 1+ | 12 (28.6) | 15 (28.8) | | |
| Missing | 14 (33.3) | 12 (23.1) | | |
| Gleason Score (%) | | | 0.021 | 0.774 |
| 10 | 1 (2.4) | 12 (23.1) | | |
| 9 | 23 (54.8) | 18 (34.6) | | |
| 8 | 7 (16.7) | 12 (23.1) | | |
| 7 | 2 (4.8) | 4 (7.7) | | |
| Missing | 9 (21.4) | 6 (11.5) | | |
| Gleason Group (%) | | | 0.335 | 0.308 |
| 9 or higher | 24 (57.1) | 30 (57.7) | | |
| <9 | 9 (21.4) | 16 (30.8) | | |
| Missing | 9 (21.4) | 6 (11.5) | | |
| Race/Ethnicity (%) | | | 0.895 | 0.162 |
| White | 20 (47.6) | 24 (46.2) | | |
| Black or African American | 11 (26.2) | 11 (21.2) | | |
| Hispanic/Latino | 3 (7.1) | 5 (9.6) | | |
| Other | 8 (19.0) | 12 (23.1) | | |
| Metastasis Burden (%) | | | 0.986 | 0.034 |
| High burden | 30 (71.4) | 37 (71.2) | | |
| Low burden | 11 (26.2) | 14 (26.9) | | |
| Uncertain | 1 (2.4) | 1 (1.9) | | |
| Visceral Mets | | | 0.931 | 0.082 |
| No | 36 (85.7) | 46 (88.5) | | |
| Yes | 6 (14.3) | 6 (11.5) | | |

SMD–Standardized Mean Difference

high metastasis burden in each cohort had visceral metastasis. Hence the cohorts were well matched for the variables include in the study (Table 1). The patient selection process is presented in S2 Fig.

## Docetaxel treatment course

In the docetaxel cohort, 47 of 52 (90.4%) patients began treatment at the standard 75 mg/m2 dosing; 10 patients had subsequent dose reduction to 60 mg/m2. The remainder of patients, 5/52 (9.6%), started at a reduced dose of 60 mg/m2; 1 patient required a dose reduction to 45 mg/m2 (S1A Table). The adverse effects leading to dose reduction in these 11 patients included: fatigue (3), neuropathy (3), febrile neutropenia (3), infection (1), and bradycardia (1) (S1B Table).

Forty-five patients (86.5%) completed 6 cycles of treatment. Six patients (11.5%) terminated docetaxel early due to adverse effects, most commonly leg edema, fatigue, LFT abnormalities, anaphylactic infusion reaction. One patient (1.9%) had docetaxel terminated before 6 cycles and moved to 2nd line treatment due to refractory disease.

Forty-two docetaxel-treated patients (82.7%) initiated subsequent therapy during the follow-up period. The most common subsequent therapy was an NHA such as abiraterone (14/52, 26.9%) or enzalutamide (11/52, 21.2%). Eight (15.4%) patients enrolled in clinical trial following docetaxel therapy. Other agents included bicalutamide (3), sipuleucel-T (3), cabazitaxel (2), and apalutamide (2) (S2 Table).

## Novel hormonal agent (NHA) treatment course

In the NHA cohort, 40 of 42 (95.2%) patients received abiraterone, and 2 of 42 (4.8%) received enzalutamide. All patients were started at each agent's respective approved dosing: abiraterone 1000 mg daily with prednisone 5 mg daily, or enzalutamide 160 mg daily. Dose reductions occurred in 4 (9.5%) patients, primarily due to abnormal LFTs (2) or hot flashes (1). (S3 Table)

Of the 42 patients treated with upfront NHA, 20 (47.6%) patients transitioned to 2nd line therapy during the follow-up period. 10% (2/20) of patients who moved onto 2nd line therapy did so within first 3 months of 1st line therapy due to rising PSA levels (S4 Table). The most common subsequent therapy following upfront NHA was an alternative NHA, e.g. enzalutamide (13/20, 65%). Three (15%) patients received docetaxel, one patient each received radium-223 (5%), olaparib (5%), Sipuleucel-T (5%) and bicalutamide (5%) (S4 Table).

## Time of next treatment

Patients who received upfront NHA had a significantly longer median TTNT of 20.7 months (95% CI = 17.0-NE [not estimable]) compared to 10.1 months (95% CI 8.75–18.2) in docetaxel treated patients (p = 0.023) (Fig 1A). Single and multivariable analyses are presented in Table 2. NHA, low metastasis burden, and lower baseline PSA levels are all associated with longer TTNT. Variables with pvalues<0.15 were retained for the multivariable regression model. In the multi-variable model adjusting for age, baseline PSA, and metastasis burden, Gleason score, docetaxel was associated with shorter TTNT compared to NHA (HR 2.15, 95% CI 1.19 −3.89, p = 0.011) (Table 2). None of the other variables remain statistically significant, but age below 65 at time of diagnosis, high baseline PSA, high Gleason score and high metastatic burden are still modestly associated with shorter TTNT, irrespective of treatment received (Table 2). However, OS was not different between the two treatment groups (p = 0.773) (S3 Fig). These findings from our real-world analysis replicate findings from clinical trial [11].

Clinical trials have suggested that high metastatic disease burden is predictive for clinical benefit from upfront treatment strategies for mHSPC [1, 2, 11]. We sought to determine whether there was evidence for heterogeneous treatment effects based on disease burden. Patients with high metastasis burden had a significantly longer TTNT with NHAs than docetaxel (25.12 [95% CI 17.03-NE] vs. 9.63 [95% CI 8.38–14.53] months, p = 0.014) (Fig 2A), while there was no significant difference in TTNT among patients with low metastasis burden (NHA 20.71 [95% CI 12.92-NE] months vs. Docetaxel 26.5 [95% CI 8.65-NE] months, p = 0.9) (Fig 2B). However, in a cox regression model predicting TTNT, the interaction between upfront therapy and metastasis burden was not significant (p = 0.28).

We further analyzed the prognostic implications of metastasis burden in the combined mHSPC cohort and in the individual docetaxel and NHA cohorts separately. In the combined cohort of docetaxel and NHA treated patients, a high metastatic burden of disease was associated with a shorter median TTNT as compared to patients with low burden of disease, 11.57

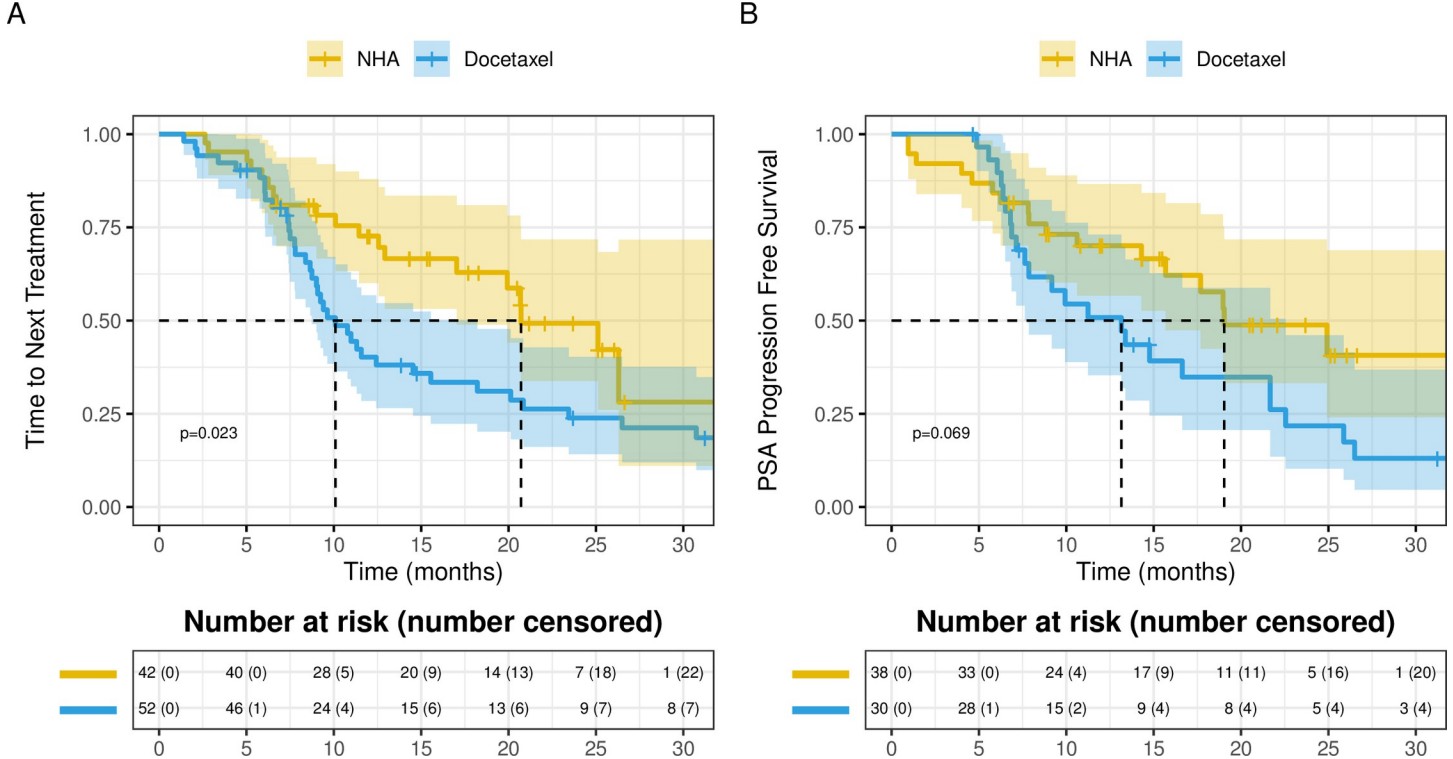

**Fig 1.** Kaplan-Meier analysis comparing mHSPC patients treated with upfront docetaxel versus novel hormonal therapy (NHA) using **A.** Failure-free survival and **B.** PSA Progression free survival.

[95% CI 9.24–20.12] vs. 20.71 [95% CI 12.92-NE] months (p = 0.034) (Fig 2C). In the docetaxel group, high burden disease is associated with shorter median TTNT as well, 9.63 [95% CI 8.38–14.53] vs. 26.5 [95% CI 8.65-NE] months (p = 0.026) (Fig 2D). However, metastasis burden was not associated with a significant TTNT difference for patients that received NHA (25.12 [95% CI 17.03-NE] vs. 20.71 [95% CI 12.92-NE] months, p = 0.79, Fig 2E).

## PSA progression

Since time to next treatment may be influenced by decisions to change therapy due to toxicity rather than disease progression, we also evaluated PSA progression between the two treatment groups. Of note, 7 of 52 docetaxel-treated patients were started on a subsequent line of therapy prior to completing 6 cycles of docetaxel treatment (due to toxicity [6 patients] or refractory disease [1 patient], as mentioned above); PSA progression after docetaxel alone could not be evaluated in these patients. Among evaluable patients, we found that those treated with docetaxel were more likely to start a subsequent therapy in the context of rising PSA levels prior to meeting PCWG criteria for PSA progression (14/45 vs. 4/42, p = 0.02). The median PSA difference between PSA nadir and PSA at the start of 2nd treatment was 0.68 vs 1.58 ng/ml among those treated with docetaxel vs NHAs (p = 0.55).

The co-variants were reasonably well matched between the remaining 30 docetaxel-treated patients and 38 NHA-treated patients with respect to age, race, log PSA at time of diagnosis, Gleason score, ECOG score, and metastasis burden (S5 Table). Among these patients, there was a trend towards longer PSA PFS with NHA compared to docetaxel, though the difference was not statistically significant (19.0 [95% CI 15.6-NE] vs. 13.2 months [95% CI 7.7–22.6],

**Table 2. Univariable and multivariable analysis of TTNT comparing upfront docetaxel to upfront NHA and the other co-variants.**

| | | Single Variable | | Multivariable | |
|---|---|---|---|---|---|
| Category | n | HR (95% CI) | pvalue | HR (95% CI) | pvalue |
| Treatment | | | | | |
| NHA | 41 | Ref | | Ref | |
| Docetaxel | 51 | 1.83 (1.05–3.19) | **0.03403** | 2.15 (1.19–3.89) | **0.01124** |
| logPSA | 92 | 1.47 (1.11–1.95) | **0.00803** | 1.28 (0.93–1.75) | 0.12989 |
| Age Category | | | | | |
| 65 and Over | 53 | Ref | | Ref | |
| Under 65 | 39 | 1.50 (0.9–2.51) | 0.1199 | 1.60 (0.95–2.70) | 0.07886 |
| ECOG Score | | | | | |
| 0 | 40 | Ref | | | |
| 1+ | 27 | 1.49 (0.84–2.63) | 0.17547 | | |
| Missing | 25 | 0.63 (0.30–1.35) | 0.23649 | | |
| Gleason Group | | | | | |
| <9 | 24 | Ref | | Ref | |
| 9 or higher | 53 | 1.84 (0.96–3.51) | 0.06516 | 1.68 (0.87–3.24) | 0.12096 |
| Missing | 15 | 1.88 (0.76–4.62) | 0.17090 | 2.22 (0.85–5.80) | 0.10223 |
| Metastasis Burden | | | | | |
| High burden | 67 | Ref | | Ref | |
| Low burden | 25 | 0.52 (0.28–0.96) | **0.03728** | 0.59 (0.31–1.13) | 0.10973 |
| Race/Ethnicity | | | | | |
| White | 43 | Ref | | | |
| Black or African American | 22 | 1.28 (0.66–2.49) | 0.46077 | | |
| Hispanic/Latino | 8 | 1.37 (0.60–3.16) | 0.45469 | | |
| Other | 19 | 0.88 (0.45–1.73) | 0.70759 | | |

p = 0.069) (Fig 1B). S6 Table presents the results for cox regression for both the univariable and multivariable analyses. In multi-variable analysis, treatment (docetaxel vs. NHA) was independently associated with PSA PFS (HR 2.06, 95% CI 1.04–4.08; p = .0039) (S6 Table). Higher log PSA at time of diagnosis was also independently associated with worse PSA PFS (S6 Table).

## Post-treatment PSA as prognostic marker in de novo mHSPC

Prior to the advent of combination therapies for *de novo* mHSPC, when such patients were treated with ADT alone, post-ADT PSA levels were shown to be independent predictors of overall survival [12]. It is not known whether post-treatment PSA also holds prognostic value in the context of modern combination treatments.

To evaluate the prognostic significance of post-treatment PSA, we examined TTNT stratified by post-treatment PSA in the overall study cohort, as well as in the individual docetaxel and NHA cohorts. Patients were stratified into 3 groups based on post-treatment PSA: ≤0.2, 0.2–4, and ≥4. In the combined docetaxel and NHA cohorts evaluable for PSA response, patients with a post-treatment PSA ≤0.2 had the longest TTNT (58.95 [95% CI 20.71-NE] vs. 11.57 [95% CI 9.63–25.12] vs. 9.40 [95% CI 7.76–20.84] months, p<0.001) (Fig 3A). Similar outcomes were observed in the individual docetaxel and NHA cohorts (Fig 3B and 3C).

In a multivariable analysis of the combined cohort, higher post-treatment PSA was independently associated with worse TTNT (0.2–4 vs. ≤0.2 HR 5.06, 95% CI 1.98–12.97; ≥4 vs. ≤0.2 HR 6.42, 95% CI 2.43–16.95) (Table 3). In the NHA cohort, similar analysis

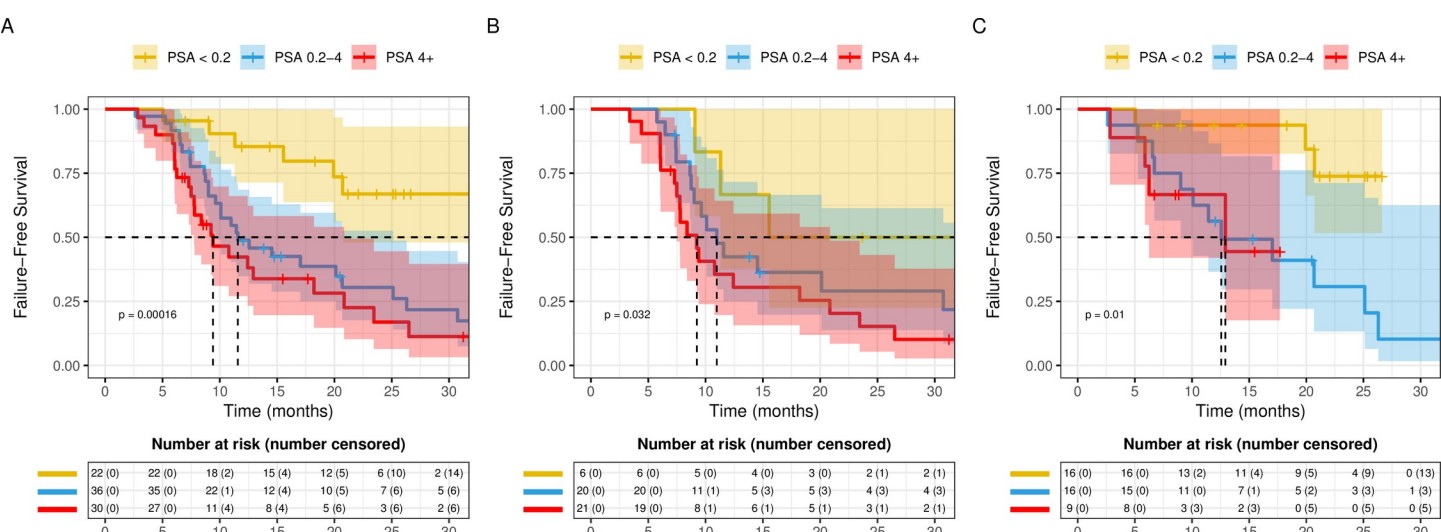

**Fig 2.** Kaplan-Meier analysis for TTNT comparing patients treated with upfront docetaxel versus upfront NHA in **A.** High metastasis **B.** Low metastasis burden patients. Kaplan-Meier analysis for time to next treatment (TTNT) comparing patients with high and low metastasis burden in **C.** all mHSPC patients, **D.** Patients treated with upfront docetaxel and **E.** Patients treated with upfront NHA.

**Fig 3.** Kaplan-Meier analysis for time to next treatment (TTNT) comparing patients stratified by drop in PSA after treatment as mentioned in methods in **A.** All mHSPC patients; **B.** upfront Docetaxel; **C.** upfront NHA treated patients.

**Table 3. Univariable and multivariable analysis of TTNT in mHSPC patients stratified by post treatment PSA and other co-variants.**

| Variable | n | Single Variable HR (95% CI) | pvalue | Multivariable HR (95% CI) | pvalue |
|---|---|---|---|---|---|
| PSA_Category | | | | | |
| PSA < 0.2 | 21 | Ref | | Ref | |
| PSA 0.2–4 | 36 | 4.11 (1.69–9.95) | **0.00176** | 5.06 (1.98–12.97) | **0.00072** |
| PSA 4+ | 29 | 5.63 (2.25–14.11) | **0.00022** | 6.42 (2.43–16.95) | **0.00018** |
| Age_Category | | | | | |
| 65 and Over | 53 | Ref | | Ref | |
| Under 65 | 39 | 1.73 (1.02–2.92) | **0.04037** | 2.4 (1.35–4.25) | **0.00278** |
| ECOG_Score | | | | | |
| 0 | 40 | Ref | | | |
| 1+ | 27 | 1.44 (0.81–2.57) | 0.2189 | | |
| Missing | 25 | 0.52 (0.23–1.2) | 0.12431 | | |
| Metastatic_Burden | | | | | |
| High burden | 67 | Ref | | Ref | |
| Low burden | 25 | 0.49 (0.26–0.92) | **0.02558** | 0.72 (0.37–1.4) | 0.33595 |
| RaceEthnicity | | | | | |
| White | 43 | Ref | | | |
| Black or African American | 22 | 1.22 (0.6–2.47) | 0.58342 | | |
| Hispanic/Latino | 8 | 1.42 (0.61–3.28) | 0.41124 | | |
| Other | 19 | 0.91 (0.46–1.8) | 0.79146 | | |

demonstrated that the PSA 0.2–4 and PSA ≥4 groups have significantly worse outcomes compared to the patients in the PSA ≤0.2 group (HR 6.01, 95% CI 1.54 − 23.44; HR 6.83, 95% CI 1.33–35.09) (S7 Table). For docetaxel, there is a trend towards worse outcomes among those with higher post-treatment PSA levels, (0.2–4 vs. ≤0.2 HR 3.42, 95% CI 0.84–13.88, ≥4 vs. ≤0.2 HR 4.35, 95% CI 1.02–18.55) with the outcome between post-treatment PSA levels ≥4 vs. ≤0.2 being statistically significant (p = 0.046) (S7 Table).

## Discussion

The treatment paradigm for mHSPC has rapidly shifted over the past 5 years from ADT monotherapy to combinations of ADT with docetaxel or NHAs such as abiraterone, enzalutamide, and apalutamide. However, data comparing these treatment options is limited. Network meta-analyses have addressed the question indirectly using cross-trial comparisons, finding no difference in overall survival between docetaxel and NHAs [6–8]. An opportunistic analysis from the STAMPEDE trial with limited power found improved failure-free survival and progression-free survival with abiraterone versus docetaxel, but no difference in overall survival [11].

Observational data may improve our understanding of the real-world use and outcomes of these treatments. We therefore conducted this study to evaluate treatment of *de novo* mHSPC patients. To our knowledge, this is the first observational analysis to directly address this question.

### Time to next treatment and metastasis burden

Upfront NHA was associated with longer median TTNT than docetaxel and remained an independent predictor controlling for demographic and clinical factors, however there was no

change in OS. This corresponds with findings from STAMPEDE, where the NHA cohort also showed a significantly longer FFS than the docetaxel cohort [11].

Time to next treatment was shorter with docetaxel than NHAs. Several factors may have contributed to this observation. First, more docetaxel-treated patients changed therapy due to toxicity (6 vs. 0, p = 0.022). Second, there appeared to be a lower clinical threshold for docetaxel-treated patients to be initiated on a 2nd line therapy compared to NHA-treated patients. For example, 14 docetaxel-treated patients were started on a 2nd line therapy before meeting formal criteria for PSA progression, compared to only 4 of NHA-treated patients (p = 0.03). In addition, the median increase in PSA from nadir at the time of 2nd line treatment initiation was 0.68 ng/ml among docetaxel-treated patients compared to 1.58 ng/ml among NHA treated patients; this difference was not statistically significant (p = 0.55), but suggests that clinicians were more inclined to initiate 2nd line therapy in docetaxel-treated patients than NHA-treated patients. The fact that docetaxel treatment is limited to 6 cycles whereas NHAs are taken on an ongoing basis may influence clinicians' response to rising PSA levels.

High burden of metastatic disease has been associated with greater overall survival (OS) benefit from upfront docetaxel in post-hoc analyses of clinical trials. In our TTNT analysis, NHAs were associated with median TTNT of over 20 months regardless of disease burden (high: 25.12 vs. low: 20.71 months) while median TTNT after docetaxel was significantly shorter in patients with high rather than low metastatic burden (9.63 vs. 26.5 months, p = 0.014). These observations raise the possibility of treatment effect heterogeneity between docetaxel and NHAs by metastatic burden. We assessed the interaction term, which was not significant (p = 0.28), but this analysis was limited by sample size. Thus, while NHAs were associated with longer TTNT in high-burden patients, it does not seem to translate into an OS benefit.

## Second-line therapy

At the end of follow-up, 17.3%, of patients remained on ADT alone after completion of docetaxel, whereas 52.4% of the NHA cohort remained on 1st line therapy. This was due in part to earlier approval for docetaxel in this setting and, consequently, longer follow-up in the docetaxel cohort (maximum of 64 months) compared to the NHA cohort (maximum of 28 months). The apparent lower clinical threshold to initiate 2nd line treatment among docetaxel-treated patients, described above, may also have contributed.

Patterns of toxicity leading to dose reductions of treatment drug reflect otherwise well-described adverse effect profiles. Most patients, 86.5%, were able to successfully complete 6 cycles of docetaxel, with the main reasons for early termination being fatigue, peripheral edema, and LFT abnormalities. Dose reductions in docetaxel were recommended for fatigue, neuropathy, and febrile neutropenia. Meanwhile, no clear cases of early termination of an NHA was observed, and dose reductions occurred only in a minority of patients, primarily in response to abnormal LFT's or hot flashes.

The prevailing practice pattern following upfront docetaxel was to transition to an NHA, with a relatively even split between abiraterone and enzalutamide. Following upfront NHA, most practitioners opted to transition to a second NHA. However, understanding that sequencing two different AR-pathway targeting agents back to back is usually associated with relatively low rates of response [13], more recent studies have explored transitioning to chemotherapy [14], radium, or targeted therapy with PARP inhibitor [15].

## PSA as a prognostic indicator

In the pre-CHAARTED era, where standard of care for mHSPC was ADT alone, post-ADT induction PSA level was shown to have strong prognostic significance, with lower PSA levels

associated with better survival outcomes [12]. To our knowledge, there is no data examining the prognostic value of post-treatment PSA in the context of modern combination therapies for mHSPC. In the overall study cohort, we observed that patients with a post-treatment PSA of ≤0.2 enjoyed the longest TTNT, whereas those with PSA 0.2–4 and ≥4 were associated with significantly shorter TTNT. This suggests that the post-treatment PSA after combination treatment in mHSPC patients has prognostic value, though further evaluation of its association with overall survival is necessary.

### Limitations

There are inherent limitations in our data given the retrospective nature of this study. Comparisons between the docetaxel and NHA-treated patients may be confounded by indication. We were unable to perform propensity-score matching due to limited sample size. However, baseline characteristics were not significantly different between the two groups.

Enzalutamide and apalutamide, which were recently approved for mHSPC, were not well-represented in our dataset. Future studies should address these newer agents as experience with them accrues.

Time to next treatment may be affected by several factors including frequency of follow-up, toxicity, and patient or provider-preference. We provided information on the decision to change treatment where available and found that most treatment changes were prompted by PSA progression. We conducted a separate PSA PFS analysis among a subset of evaluable patients which showed a trend towards longer PSA PFS with NHAs compared to docetaxel. Larger observational cohorts or prospective data are needed to compare long-term survival outcomes between these two treatment strategies for mHPSC.

### Conclusion

In conclusion, this retrospective single-center study observed that upfront NHA treatment for *de novo* mHSPC was associated with longer TTNT compared to docetaxel, without impacting OS. Median TTNT was longer among high-burden patients treated with NHAs rather than docetaxel, whereas median TTNT was similar regardless of treatment in low-burden patients; the interaction was not significant. Even with the current limitation of this study, we hope this study will help clinicians council patients on what to expect when put on docetaxel vs NHA for mHSPC on subsequent treatments and journey in disease management. Data from this study can inform a discussion over the clinical outcomes of upfront NHA versus docetaxel. Patients who value time off treatment may gravitate towards docetaxel, whereas those who value a longer time to next treatment may prefer NHAs. Finally, we showed that in the context of modern combination treatments for mHSPC, post-treatment PSA levels were associated with time to next treatment, suggesting prognostic value of this marker which should be evaluated in larger, prospective datasets.

### Supporting information

**S1 Fig. A.** Sema4-automated oncology data retrieval and curation platform. **B.** Sema4 automated abstraction engine for cancer diagnosis.
(PPTX)

**S2 Fig. Flow chart demonstrating patient selection for docetaxel and NHA cohorts.**
(PPTX)

**S3 Fig. Kaplan-Meier analysis comparing mHSPC patients treated with upfront docetaxel versus novel hormonal therapy (NHA) using overall survival.**
(PDF)

**S1 Table. A.** Demonstrates dose and dose reduction for docetaxel cohort. **B.** Lists adverse effect associated with dose reduction.
(XLSX)

**S2 Table. Demonstrated the no of cycles and 2ⁿᵈ line therapy for patients treated with upfront docetaxel.**
(XLSX)

**S3 Table. Demonstrates dose and dose reduction for NHA cohort.** B. Lists adverse effect associated with dose reduction.
(XLSX)

**S4 Table. Demonstrated the 2ⁿᵈ line therapy for patients treated with upfront NHA.**
(XLSX)

**S5 Table. Table of co-variants for the docetaxel and NHA cohorts for PSA progression free survival analysis.**
(XLSX)

**S6 Table. Univariable and multivariable analysis of PSA PFS comparing upfront docetaxel to upfront NHA and other co-variants.**
(XLSX)

**S7 Table. Univariable and multivariable analysis for time to next treatment (TTNT) for upfront NHA or upfront docetaxel treated patients stratified by post treatment PSA and other co-variants.**
(XLSX)

## Author Contributions

**Conceptualization:** Bobby K. Liaw, Rong Chen, William K. Oh.

**Data curation:** Tommy Mullaney, Tony Prentice, Xiang Zhou, Eric E. Schadt.

**Formal analysis:** Sunny Guin, Kristin Ayers, Bonny Patel, Timmy O'Connell, Matthew Deitz, Michael Klein, Marc Fink.

**Writing – original draft:** Sunny Guin.

**Writing – review & editing:** Sunny Guin, Bobby K. Liaw, Tomi Jun, Kristin Ayers, Scott Newman.

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
