## [Decision Letter · Decision Letter 0]

14 Jun 2022

PONE-D-22-04683Management of de novo metastatic hormone-sensitive prostate cancer: a comprehensive report of a single-center experiencePLOS ONE

Dear Dr. Guin,

Thank you for submitting your manuscript to PLOS ONE. After careful consideration, we feel that it has merit but does not fully meet PLOS ONE’s publication criteria as it currently stands. Therefore, we invite you to submit a revised version of the manuscript that addresses the points raised during the review process.

We look forward to receiving your revised manuscript.

Kind regards,

Isaac Yi Kim, MD, PhD, MBA

Academic Editor

PLOS ONE

Journal Requirements:

Reviewers' comments:

Reviewer's Responses to Questions

**Comments to the Author**

1. Is the manuscript technically sound, and do the data support the conclusions?

Reviewer #1: Yes

Reviewer #2: Yes

2. Has the statistical analysis been performed appropriately and rigorously? 

Reviewer #1: Yes

Reviewer #2: Yes

3. Have the authors made all data underlying the findings in their manuscript fully available?

Reviewer #1: Yes

Reviewer #2: Yes

4. Is the manuscript presented in an intelligible fashion and written in standard English?

Reviewer #1: Yes

Reviewer #2: Yes

5. Review Comments to the Author

Reviewer #1: The authors conducted a database survey to identify risk factors among mCSPC patients treated with upfront chemotherapy or NHA. The primary end-point was TTNT. I have some comments as below.

1. Were there any neuroendocrine tumor in origin included in this study?

2. The statistical power is low because of the small study number.

3. The study design and the manuscript wring is robust.

Reviewer #2: The authors present a retrospective analysis of real world data from their institution - comparing outcomes for patients with metastatic castration-sensitive prostate cancer treated either with docetaxel or novel hormonal agents. The question the authors are trying to address is relevant and timely since this clinical scenario is common and there is limited data to guide the decision of proceeding with docetaxel, NHA, or now with triplet therapy (both docetaxel and NHA). They report a longer time to next treatment (TTNT) for patients treated with docetaxel compared to NHA therapy but no difference in survival.

The authors are using an automated abstraction platform, and data to demonstrate accuracy of this platform would be useful to ensure soundness of their conclusions. They refer to the figures in the supplement and say that a subset was reviewed manually for quality assurance but I would like more data regarding the accuracy of their platform.

Some caveats are required related to the interpretation of their data, which the authors also acknowledge in their discussion:

1) there may be selection bias where patients offered docetaxel have more aggressive disease than those offered NHA. Although visceral/metastatic burden variables were reported to be similar, there seems to be a difference in Gleason between the two groups and there may be other differences that were not measured. Patients offered NHA may also be less fit (fewer ECOG 0) and thus oncologists may be slower to offer more intense therapies, or slower to change therapy.

2) indication for next line of therapy may be different for patients offered docetaxel vs NHA. For example, patients with PSA progression after docetaxel may be offered NHA; while patients with PSA progression on NHA may be monitored until radiographic progression to start docetaxel. The authors state that the PSA was higher in the NHA group than the docetaxel group. The authors also mention that toxicity is different between the two types of treatment and was more often a reason for discontinuation/change of therapy in the docetaxel group.

for these reasons, the clinical significance of their endpoint is unclear to me - even if TTNT is shorter in docetaxel group, it is not necessarily a reason to favor NHA over docetaxel, especially since OS is the same in their analysis.

The authors state that patients started on a next line of therapy before PSA progression were excluded - how many patients and what types of therapies? Were there more in the docetaxel vs the NHA group?

Were there differences in subsequent lines of therapy for docetaxel vs NHA?

Some other comments:

line 51 in abstract - clarify that this is PSA nadir

line 74 - would add STAMPEDE to body of evidence supporting treatment intensification in mHSPC.

In Table 1:

why is PSA being reported as log10PSA?

Why is Gleason 10 presented before 7, 8, and 9?

It is somewhat unusual to present both mean and categories of variables such as age and Gleason in Table 1; or mean plus log10 as for PSA.

Line 389 - needs attention to correct grammar

6. PLOS authors have the option to publish the peer review history of their article (what does this mean?). If published, this will include your full peer review and any attached files.

Reviewer #1: No

Reviewer #2: No

---

## [Author Response · Author response to Decision Letter 0]

21 Jul 2022

Isaac Yi Kim, MD, PhD, MBA

Academic Editor

PLOS ONE

July 20th 2022

Dear Dr. Kim, 

Thank you for the feedback and insightful comments from the editorial office and the reviewers. We appreciate the opportunity to revise the manuscript. We accordingly revised the text, figures and tables, and supplementary content. We have made changes to the manuscript per PLOS ONE style requirement. Changes in the main text are highlighted. Below are the detailed point-by-point responses to the reviewers’ comments.

Reviewer 1

Comment 1: 

Were there any neuroendocrine tumor in origin included in this study?

Answer: None of the patients in the final cohort had neuroendocrine histology.

Comment 2: 

The statistical power is low because of the small study number.

Answer: We understand the reviewer concern. We have been rigorous with our methods and have highlighted limitations of our study including sample size.

Comment 3: 

The study design and the manuscript wring is robust.

Answer: We thank the reviewer for appreciating our study design and manuscript writing.

Reviewer 2: 

Comment 1:

The authors present a retrospective analysis of real world data from their institution - comparing outcomes for patients with metastatic castration-sensitive prostate cancer treated either with docetaxel or novel hormonal agents. The question the authors are trying to address is relevant and timely since this clinical scenario is common and there is limited data to guide the decision of proceeding with docetaxel, NHA, or now with triplet therapy (both docetaxel and NHA). They report a longer time to next treatment (TTNT) for patients treated with docetaxel compared to NHA therapy but no difference in survival.

The authors are using an automated abstraction platform, and data to demonstrate accuracy of this platform would be useful to ensure soundness of their conclusions. They refer to the figures in the supplement and say that a subset was reviewed manually for quality assurance but I would like more data regarding the accuracy of their platform.

Answer: We understand the reviewer concern. We have referred to a recently presented abstract in American Medical Informatics Association (AMIA) Informatics Summit in 2022 highlights the efficiency of diagnosis extraction by the abstraction platform highlighted in Supplemental Figure 1. 

Besides, once a cohort is selected for a study, we carry out manual chart review of patient records to improve on automated abstraction for final analysis. It is clearly stated in Methods subsection “Patient Cohort Identification” in the text-

 “After the cohorts were identified, chart review was carried out to confirm duration of upfront docetaxel or NHA therapy, adverse events, identification of 2nd line therapy, and start date of 2nd line therapy. Additional clinical and demographic variables such as PSA values, age, sex, race, performance status, Gleason score, and metastasis burden at time of diagnosis were also evaluated.”

Comment 2: 

there may be selection bias where patients offered docetaxel have more aggressive disease than those offered NHA. Although visceral/metastatic burden variables were reported to be similar, there seems to be a difference in Gleason between the two groups and there may be other differences that were not measured. Patients offered NHA may also be less fit (fewer ECOG 0) and thus oncologists may be slower to offer more intense therapies, or slower to change therapy.

Answer: We understand the reviewer concern. We agree treatment change in real world scenario can be attributed to several factors. We explain the reasons for the difference in time to next treatment in Discussion in the text 

“Time to next treatment was shorter with docetaxel than NHAs. Several factors may have contributed to this observation. First, more docetaxel-treated patients changed therapy due to toxicity (6 vs. 0, p = 0.022). Second, there appeared to be a lower clinical threshold for docetaxel-treated patients to be initiated on a 2nd line therapy compared to NHA-treated patients. For example, 14 docetaxel-treated patients were started on a 2nd line therapy before meeting formal criteria for PSA progression, compared to only 4 of NHA-treated patients (p=0.03). In addition, the median increase in PSA from nadir at the time of 2nd line treatment initiation was 0.68 ng/ml among docetaxel-treated patients compared to 1.58 ng/ml among NHA treated patients; this difference was not statistically significant (p=0.55), but suggests that clinicians were more inclined to initiate 2nd line therapy in docetaxel-treated patients than NHA-treated patients. The fact that docetaxel treatment is limited to 6 cycles whereas NHAs are taken on an ongoing basis may influence clinicians’ response to rising PSA levels.”

Moreover, we are not claiming one treatment is better than the other. We are reporting on the treatment patterns observed in patients and try to understand the underlying reason for the observations made.

Comment 3: 

indication for next line of therapy may be different for patients offered docetaxel vs NHA. For example, patients with PSA progression after docetaxel may be offered NHA; while patients with PSA progression on NHA may be monitored until radiographic progression to start docetaxel. The authors state that the PSA was higher in the NHA group than the docetaxel group. The authors also mention that toxicity is different between the two types of treatment and was more often a reason for discontinuation/change of therapy in the docetaxel group.

for these reasons, the clinical significance of their endpoint is unclear to me - even if TTNT is shorter in docetaxel group, it is not necessarily a reason to favor NHA over docetaxel, especially since OS is the same in their analysis.

Answer: We understand the reviewer concern. We do not make a claim that one treatment is better than the other. We report on real treatment patterns for mHSPC patients and found that time on treatment is shorter on docetaxel than NHA however there is no change in OS. We acknowledge time on treatment can be influenced by many factors as discussed in the Discussion section and highlighted above. We hope this study help clinicians council patients on what to expect when put on docetaxel vs NHA for mHSPC on subsequent treatments and journey in disease management.

Comment 4: 

The authors state that patients started on a next line of therapy before PSA progression were excluded - how many patients and what types of therapies? Were there more in the docetaxel vs the NHA group?

Answer: The patients who were put on next line of therapy before meeting the PCWG criteria for PSA progression were excluded from the PSA Progression Free Survival Analysis. 14 patients receiving docetaxel were moved to next treatment before meeting PSA progression guideline compared to 4 in NHA group. This is noted in Results, subsection “PSA progression”

“Among evaluable patients, we found that those treated with docetaxel were more likely to start a subsequent therapy in the context of rising PSA levels prior to meeting PCWG criteria for PSA progression (14/45 vs. 4/42, p=0.02).”

Comment 5: 

Were there differences in subsequent lines of therapy for docetaxel vs NHA?

Answer: Supplemental Table 2 and 4 list the subsequent line of therapy after docetaxel or NHA. The findings are described in Results under subsection “Docetaxel Treatment Course” and “Novel Hormonal Agent Treatment Course”. NHA were the most common drugs used after docetaxel or NHA. 

Comment 6:

 line 51 in abstract - clarify that this is PSA nadir

Answer: Thank you for the suggestion. We have clarified this in the Abstract now by adding- 

“lower post-treatment PSA levels (PSA nadir)”.

Comment 7: 

line 74 - would add STAMPEDE to body of evidence supporting treatment intensification in mHSPC.

Answer: Line 74 talks about the two landmark trials CHAATED and LATIDUTE that lead to use of docetaxel and abiraterone to standard of care for mHSPC patients in USA. Since STAMPEDE do not fit that criterion, it is not discussed in the Introduction. However, we discuss results from STAMPEDE trials extensively in Discussion.

Comment 8: 

In Table 1:

why is PSA being reported as log10PSA?

Why is Gleason 10 presented before 7, 8, and 9?

It is somewhat unusual to present both mean and categories of variables such as age and Gleason in Table 1; or mean plus log10 as for PSA.

Answer: Thank you for this comment. We have reported log10PSA because absolute PSA values have a heavy right tail containing outliers with very large values which can skew the mean and the results in regression models. Taking the log changes the distribution to more closely resemble a normal distribution. Moreover, we have used mean and categories for same variables as age; mean and log10 for PSA to analyze the variables in detail to understand how they match between cohorts. Sometimes two cohorts can have a similar mean for a continuous variable but the categories can be distributed quite differently. We have fixed the order of Gleason scores in Table 1.

Comment 9: 

Line 389 - needs attention to correct grammar

Answer: We corrected the grammar. The sentence reads-

 “this study will help clinicians….”

Sincerely,

William Oh, MD

---

## [Editor Report · Decision Letter 1]

4 Aug 2022

Management of de novo metastatic hormone-sensitive prostate cancer: a comprehensive report of a single-center experience

PONE-D-22-04683R1

Dear Dr. Oh,

We’re pleased to inform you that your manuscript has been judged scientifically suitable for publication and will be formally accepted for publication once it meets all outstanding technical requirements.

Kind regards,

Isaac Yi Kim, MD, PhD, MBA

Academic Editor

PLOS ONE
---

## [Editor Report · Acceptance letter]

11 Aug 2022

PONE-D-22-04683R1 

Management of *de novo* metastatic hormone-sensitive prostate cancer: a comprehensive report of a single-center experience 

Dear Dr. Oh:

I'm pleased to inform you that your manuscript has been deemed suitable for publication in PLOS ONE. Congratulations! Your manuscript is now with our production department. 

Kind regards, 

on behalf of

Dr. Isaac Yi Kim 

Academic Editor

PLOS ONE